# Hyperspectral Target Detection with an Auxiliary Generative Adversarial Network

**Yanlong Gao** , **Yan Feng \*** and **Xumin Yu**

School of Electronics and Information, Northwestern Polytechnical University, Xi'an 710129, China; gordonsj@mail.nwpu.edu.cn (Y.G.); xuminyu@mail.nwpu.edu.cn (X.Y.)
* Correspondence: sycfy@nwpu.edu.cn

**Abstract:** In recent years, the deep neural network has shown a strong presence in classification tasks and its effectiveness has been well proved. However, the framework of DNN usually requires a large number of samples. Compared to the training sets in classification tasks, the training sets for the target detection of hyperspectral images may only include a few target spectra which are quite limited and precious. The insufficient labeled samples make the DNN-based hyperspectral target detection task a challenging problem. To address this problem, we propose a hyperspectral target detection approach with an auxiliary generative adversarial network. Specifically, the training set is first expanded by generating simulated target spectra and background spectra using the generative adversarial network. Then, a classifier which is highly associated with the discriminator of the generative adversarial network is trained based on the real and the generated spectra. Finally, in order to further suppress the background, guided filters are utilized to improve the smoothness and robustness of the detection results. Experiments conducted on real hyperspectral images show the proposed approach is able to perform more efficiently and accurately compared to other target detection approaches.

**Keywords:** hyperspectral images; target detection; generative adversarial network



## 1. Introduction

Hyperspectral images (HSIs) obtained by remote sensing systems usually contain hundreds of continuous narrow bands over a wide range of an electromagnetic spectrum. Each pixel of HSIs forms a continuous spectrum, which is able to provide abundant useful spectral information that is beneficial to distinguish different objects [1]. By exploiting the different electromagnetic spectra of materials, targets of interest in the hyperspectral scene can be detected and identified through effective processing algorithms.

As one of the most important research areas, target detection (TD) is basically a binary classifier which aims at separating target pixels from the background with known target spectra. Furthermore, just as fields of synthetic aperture radar (SAR) target recognition [2], electromagnetic sources recognition [3] and single-image super-resolution [4], which are very likely to be confronted with the problem of insufficient training samples, TD tasks also face the challenge of a small size of labeled training samples. Because of the small fraction of pixels being labeled as targets, the method of statistical hypothesis testing is usually exploited in classical target detection algorithms. Representative algorithms include the linear spectral matched filter (SMF), matched subspace detector (MSD) and adaptive coherence/cosine estimator (ACE). These algorithms define the target spectral characteristics by either a single target spectrum or a target subspace, while modeling the background statistically by a Gaussian distribution or a subspace representing either the whole or the local background statistics. Through the generalized likelihood ratio test (GLRT), the output for a certain pixel is obtained [1]. As kernel methods in machine learning are introduced to target detection, the approaches mentioned above are further extended to nonlinear versions, such as kernel SMF and kernel MSD [5], which can solve nonlinear

problems to some extent. However, how to design a suitable kernel remains a problem. Constrained energy minimization (CEM), as a linear filter, is another target detection approach which constrains a desired target signature while being able to minimize the total energy of the output of other known signatures [6].

Meanwhile, target detection algorithms based on sparse representation are also extensively explored by researchers. Central to sparse representation is the idea that a signal can be expressed as a linear combination of very few atoms of an over-complete dictionary consisting of a set of training samples from all classes. Then, the class information can be revealed by the sparse coefficients under the assumption that signals from different classes lie in different spaces [7]. Typical approaches include the original sparse representation-based target detector (STD) and its variants: sparse representation-based binary hypothesis detector (SRBBH) [8] and sparse and dense hybrid representation-based target detector (SDRD) [9]. On the basis of STD, SRBBH perceives the detection problem as a competition between the background-only subspace and the target-and-background subspace under two different hypotheses. However, different from SRBBH, SDRD emphasizes that the background and the target sub-dictionaries collaboratively compete with each other, which facilitates the learning of sparse and dense hybrid representations for the test pixels.

Although progress has been achieved and many problems in detection tasks have been solved, challenges are still there within the realm of hyperspectral target detection. First, due to sensor noise and other factors, the target spectrum of the same material may vary dramatically, which makes it hard to represent the target spectrum using only a single spectrum or a subspace. Further, since spatial resolutions of HSIs are usually lower than that of multi-spectral and chromatic images, mixing pixels covering several materials still exist, which makes detection tasks much more complicated. Moreover, the problem of target detection is mostly nonlinear, but the approaches mentioned earlier are more likely to make decisions under linear assumptions, which does not do justice to the complex reality of HSI processing. Furthermore, it is noteworthy that many of these approaches just fail to work well when nonlinearity increases.

Recently, the deep neural network (DNN), which has achieved great success in the field of computer vision, has attracted the attention of the HSI processing community. However, due to the small training set with very few target spectra in the task of hyperspectral detection, DNNs designed for large datasets can hardly be utilized directly. In order to utilize the great representation power of a DNN, the training strategy, as well as the network structure and function must be carefully designed.

Furthermore, based on the whole new avenue opened up by DNNs, approaches of [10–14] have been proposed accordingly, in which the neural network has been exploited as an unsupervised feature extractor. Specifically, in [10], a distance-constrained stacked sparse autoencoder is utilized for feature extraction, which is able to maximize the distinction between the target pixels and the other background pixels in the feature space, and the process is followed by background suppression via a simple detector based on a radial basis function kernel. The authors of Ref. [11] introduce two adversarial losses in the autoencoder to reconstruct enhanced hyperspectral images and utilize the RX detector for anomaly detection. Meanwhile, the autoencoder also proves effective to be exploited to extract spectral features [12] which are further processed with band selection and spatial post-processing techniques for background suppression. Similar processing techniques can be found in deep latent spectral representation learning proposed in [13]. Another approach that is comparable is 3-D macro–micro residual autoencoder (3-D-MMRAE) in which a convolutional autoencoder is built to extract the spectral and spatial features from both the macro and the micro branches, and the extracted features are further fed to a hierarchical radial basis function (hRBF) detector for target preservation and background suppression [14]. The results have demonstrated the effectiveness of these approaches in HSI processing, but the underlying mechanism of the transformation process using an autoencoder remains to be uncovered.

Besides unsupervised approaches, supervised approaches have also garnered much attention. Furthermore, among such approaches, the subtraction pixel pair feature (SPPF), which tries to enlarge the training set by using pairs between target spectra and background spectra [15], eclipses many others as a very effective approach. To be specific, the approach acquires a sufficiently large number of samples using SPPF and ensures the excellent performance of the multi-layer convolutional neural network. Perhaps more effective is the approach of [16], which combines the representation-based method together with the linear mixing model to generate adequate background and target samples to train the neural network. Whilst being compared to [15], the subtraction process in [16] is performed in the final layer rather than the input layer, which improves the detection performance.

Apart from the approaches mentioned above, efforts performed from other angles which prove fruitful are [17–19]. The authors of Ref. [17] utilize a semi-supervised learning method through a two-step training process, which unfolds with the unsupervised pre-training of a generative adversarial network, followed by the fine-tuning of a discriminator through limited labeled samples. The authors of Ref. [18] realize few shot learning based on domain adaptive approaches. To be specific, a network including both feature fusion and channel attention is built for feature extraction and, then, the learned features from different sensors are correlated through a discriminator whereby the weights learned in the source domain are able to be transferred to the target domain for the following few shot learning. The authors of Ref. [19] approach the detection problem based on multiple instance learning, in which a sparsity constraint is introduced for the estimated attention factor of the positive training bags and ensures the approach yields superior performance.

In this paper, we propose a novel hyperspectral target detection approach with an auxiliary generative adversarial network. It is a neural network which can be trained with very limited target spectra directly without using any other preprocessing approaches. Similar to the general generative adversarial network [20], the proposed network mainly contains two parts: a generator and a discriminator. The generator generates simulated target spectra and background spectra, which ease the problem of insufficient training samples. Then, the initial detection results can be acquired by a classifier which is highly correlated with the discriminator. Since the neural network only deals with the spectral information of pixels, post-processing techniques are added to make full use of the spatial information. Furthermore, in order to preserve the edges of the targets, several guided filters [21] are utilized which can further suppress the background and improve the smoothness as well as the robustness of the detection results.

Extensive experiments have been conducted on real HSIs and the results show the proposed approach is more effective compared to the conventional target detection approaches. The contributions of the proposed approach are summarized as follows:

(1) A novel hyperspectral target detection framework based on the generative adversarial network is proposed, which can exhibit good performance while being trained with limited target spectra.

(2) During the training process, a generator capable of generating simulated target spectra is obtained. Furthermore, the generated targets as well as the background spectra allow us to better understand the real distributions of the target and the background.

(3) A combination of guided filters is adopted to further suppress the background, which ensures the smoothness and robustness of the results and makes the processing of various sizes of targets possible.

The remainder of this article is organized as follows: Section 2 introduces the relevant theories and the proposed approach based on the generative adversarial network in detail. Section 3 shows the analysis process and the experimental results. Section 4 puts forward a brief discussion on similar approaches and sheds light upon the future work. Section 5 draws the conclusion.

## 2. Materials and Methods

Before introducing the proposed approach, we first explicate the theories related to generative adversarial network and guided filter, which were used for spectral target detection and spatial smoothing in the proposed approach, respectively. In the following section of this paper, we examined the circumstantial process.

### 2.1. Generative Adversarial Network

Generative adversarial network (GAN) was first proposed and used for estimating generative models in [20]. To achieve good performance, GAN utilizes two models: a generator and a discriminator, both of which can be realized by neural networks.

The basic idea of GAN is that the generator tries to generate fake samples and fool the discriminator, while the discriminator tries to distinguish between real samples coming from the data and fake samples coming from the generator. This adversarial competition drives both the generator and the discriminator to improve their performance until the generated or fake samples cannot be distinguished from the real ones. This process can be formulated in a more mathematical manner.

Suppose $p_{data}(x)$, $p_z(z)$ denote data distribution and a random noise distribution, respectively. The generator builds a mapping $p_z(z) : G(z; \theta_g) \to p_g(x)$, where $p_g(x)$ denotes the distribution of generated samples. As for the discriminator, let $D(x; \theta_d)$ denote the probability of $x$ coming from $p_{data}(x)$ rather than $p_g(x)$. Then, the objective function of the generator and the discriminator, denoted as $L_g$ and $L_d$, can be formulated as Equations (1) and (2), respectively.

$$\min_G L_g = \mathbb{E}_{z \sim p_z(z)}[\log(1 - D(G(z)))] \tag{1}$$

$$\max_D L_d = \mathbb{E}_{x \sim p_{data}(x)}[\log D(x)] + \mathbb{E}_{z \sim p_z(z)}[\log(1 - D(G(z)))] \tag{2}$$

Thus, during alternate iteration of the training process, the final value function $V(D, G)$ can be formulated as Equation (3).

$$\min_G \max_D V(D, G) = \mathbb{E}_{x \sim p_{data}(x)}[\log D(x)]$$
$$+ \mathbb{E}_{z \sim p_z(z)}[\log(1 - D(G(z)))] \tag{3}$$

The value function in Equation (3) can be further interpreted as a distance metric, i.e., Jensen–Shannon divergence. In practice, however, vanilla GAN proposed in [20] is usually hard to train in deep networks. Researchers holding different viewpoints have been working hard to propose various variants which can be optimized more easily, including the activation function, objective function and network architectures, etc.

For example, in [22], several strategies for training stable deep convolutional GANs are suggested, including adopting batchnorm, using ReLU activation in the generator other than the output layer and using LeakyReLU activation in the discriminator. On the other hand, through the variational divergence estimation network, the training objective of GAN is generalized to arbitrary f-divergences in [23]. Moreover, in order to accommodate different application scenarios, modified network architectures are introduced, such as information maximizing GAN (InfoGAN) [24] and auxiliary classifier GAN (ACGAN) [25].

Generally speaking, generative adversarial network is mainly utilized either to model the distribution of samples or to generate samples. Although there exist several criteria for evaluating the quality of generated samples, open issues still remain in the particular fields.

In the proposed approach, we exploit the core idea of GAN and try to detect targets through discrimination. To be more concrete, a detection network is built with an auxiliary GAN which provides supplementary samples and shares several layers with the detector. Furthermore, to fix the problem of the detection network in being only able to deal with spectral information, a spatial post-processing technique was employed in the training process.

### 2.2. Guided Filter

Guided image filtering is one kind of self-adaptive image filters. It computes the filtering process according to the content of a guided image which can be the input image itself or another image. To put it more specifically, the guided filter is assumed to be a local linear model between the guided image $I$ and the filtering output $q$. Suppose $q$ is a linear transformation of $I$ in a local area $w_k$ centered at the pixel $k$ [21]:

$$q_i = a_k I_i + b_k, \forall i \in w_k \tag{4}$$

where $a_k$ and $b_k$ denote linear coefficients which are assumed to be constant in $w_k$. On the other hand, the output $q$ is modeled as the input $p$ which eliminates a few other parts $n$, defined as:

$$q_i = p_i - n_i \tag{5}$$

When Equations (4) and (5) are combined together, the previous tough issue is transformed into a minimization problem with a quadric formula:

$$\mathbb{E}(a_k, b_k) = \sum_{i \in w_k} \left[ (a_k I_i + b_k) - p_i \right]^2 + \epsilon a_k^2 \tag{6}$$

where $\epsilon$ is a regularization parameter which penalizes large $a_k$.

As an edge-preserving smooth operator, guided filter is faster and works better near edges when compared to the popular bilateral filters. Therefore, we adopted guided filter as a post-processing technique in the detection algorithm to further suppress the background information and the tiny isolated points in the previous detection maps.

### 2.3. Proposed Approach

In this section, we explain the proposed detection approach in detail. Firstly, a general introduction about the detection framework with circumstantial explanations was given. After that, the architectures of the network along with the training objective were introduced. Finally, a bank of guided filters used for spatial smoothing was illustrated briefly.

#### 2.3.1. General Framework

Given the fact that labeled target spectra are quite limited, unlabeled spectra including targets and backgrounds should, therefore, be utilized much more efficiently. To this end, our approach utilized the generative adversarial network to model the distributions of both the target spectra and the background spectra. Furthermore, to our delight, we found when the generative adversarial network converged, the generator would well generate both the simulated target spectra and the background spectra, and the discriminator was also able to extract useful information to be used for target detection. Hence, a target detector could be merged with the discriminator of a GAN. The general framework of the proposed approach is shown in Figure 1.

As demonstrated in Figure 1, the whole process of the proposed approach was divided into two phases: network training and detecting. In the training process, the target spectra together with the unlabeled spectra to be detected were regarded as the targets and the backgrounds, respectively, and were fed into the general adversarial network for training. Although it seemed venturesome to deem all of the unlabeled spectra as backgrounds, the generative adversarial network proved to be able to counteract the wrong apportionment very effectively with simulated samples. When the training process converged, the discriminator of GAN was selected as the detector in the detecting phase. In this process, we obtained an initial detection map through the discriminator based on the spectral information. Furthermore, to further suppress the background and the tiny outliers, guided filters were utilized to improve the smoothness and robustness of the detection results.

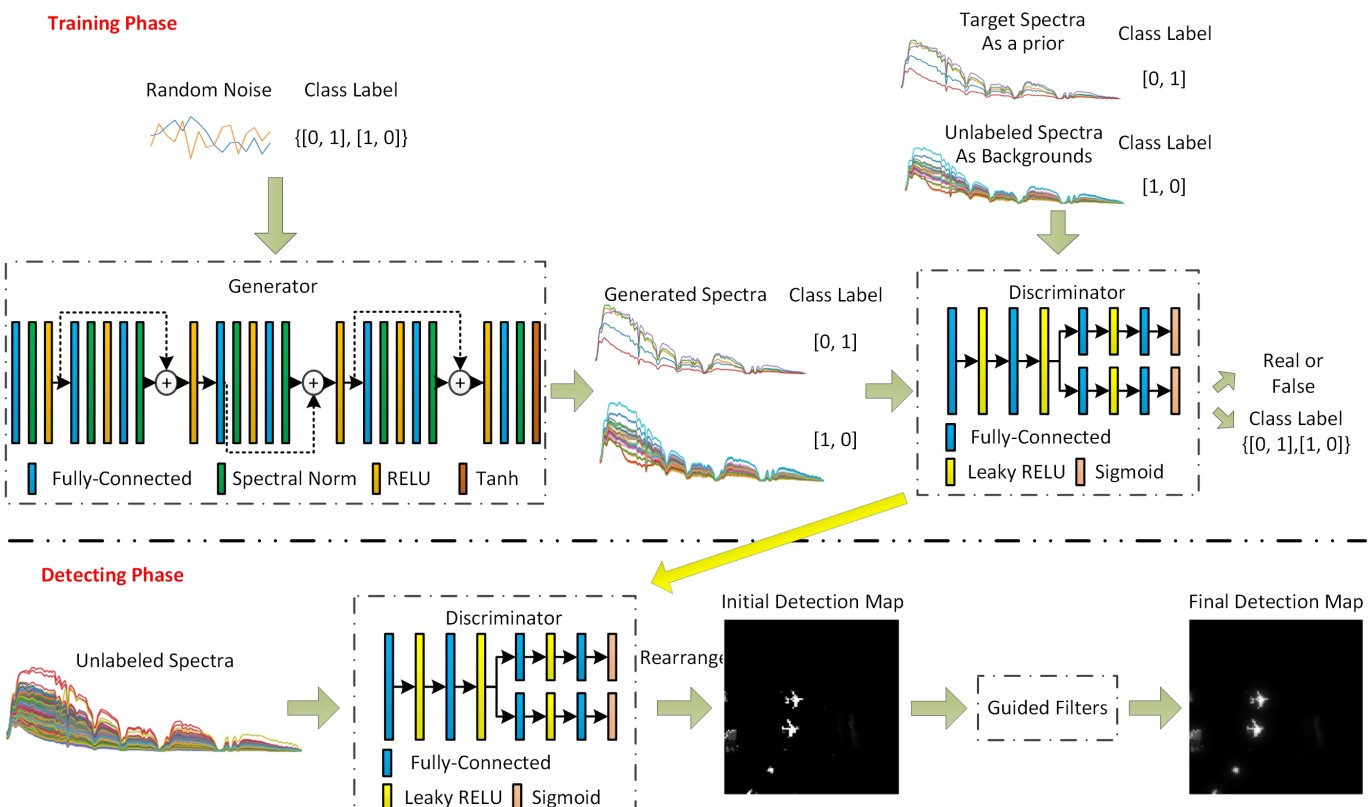

**Figure 1.** General network of our approach.

### 2.3.2. Network Architecture and Objective Function

With respect to the specific network architectures, it should be noted that the neural network during training was divided into a generator network and a discriminator network. Furthermore, we followed different criteria while building these two networks for target detection.

As shown in Figure 1, a residual structure was adopted in the generator, which was very expressive and could be used to well model the complicated distributions of the target and the background spectra. Meanwhile, in order to stabilize the training process, spectral normalization (SN) layers which were proposed in [26] were added between every two fully connected layers. As for the discriminator, we designed a dual branch structure. One branch output the truth value indicating whether the sample was real or false, and the other output the class label indicating whether the sample was target or background. The dual branches shared as many layers as possible, linking the generative adversarial network with a detector. Parameters of our network are listed in Table 1.

As the discriminator in the proposed network was able to provide two kinds of probabilities through the dual branches, viz. one for the truth value and the other for the class label, the loss of the discriminator denoted as $L_d$ could, accordingly, be represented by two parts, i.e., $L_{adv}$ for the truth loss and $L_{clc}$ for the class loss, which were similar to that of ACGAN as proposed in [25] and could be defined as Equation (7):

$$L_d = L_{adv} + L_{clc} = [\sum_i \max(0, 1 - s_r^i) + \sum_j \max(0, 1 + s_g^j)] - \sum_{i,j}[c \log \hat{c} + (1 - c) \log(1 - \hat{c})] \tag{7}$$

where $s_r^i$ and $s_g^j$ denote the truth values for real and generated samples separately and $c$ and $\hat{c}$ denote the real and the predicted class labels for all the samples, respectively. As shown

in Equation (7), the first two terms represented the hinge version of the adversarial loss as proposed in [26,27], while the third term represented the cross-entropy between the real and the predicted class labels. Through integrated loss, as the network converged, the discriminator could not only distinguish between real and generated samples, but also provide a class label for each sample.

**Table 1.** Parameter of the proposed network.

| Network | Layer | Type | Shape | Activation |
|---------|-------|------|-------|------------|
| Generator | Input | Fully Connected | $16 \times 128$ | None |
| | | SN | | ReLU |
| | Res. 1–3 | Fully Connected | $128 \times 64$ | None |
| | | SN | | ReLU |
| | | Fully Connected | $64 \times 128$ | None |
| | Output | Fully Connected | $128 \times N_{bands}$ | None |
| | | SN | | Tanh |
| Discriminator | Input | Fully Connected | $N_{bands} \times 256$ | Leaky ReLU |
| | | Fully Connected | $256 \times 128$ | Leaky ReLU |
| | Branch 1 | Fully Connected | $128 \times 32$ | Leaky ReLU |
| | | Fully Connected | $32 \times 1$ | Sigmoid |
| | Branch 2 | Fully Connected | $128 \times 32$ | Leaky ReLU |
| | | Fully Connected | $32 \times 2$ | Sigmoid |

As for the objective of the generator (denoted as $L_g$), the case became quite simple. As demonstrated in Equation (8):

$$L_g = L_{gen} + \alpha \times L_{reg} = -\sum_j s_g^j + \alpha \times \sum_p |\hat{X}_{p+1}$$
$$+\hat{X}_{p-1} - 2\hat{X}_p| \tag{8}$$

where $\hat{X}_p$ denotes a certain value on position $p$ of the generated spectrum $\hat{X}$, the first term denotes the ordinary generative loss and is denoted as $L_{gen}$, and the second term, represented as $L_{reg}$, refers to the second-order difference and denotes a relatively weak but useful penalty which could help to smoothen the generated spectra and stabilize the training process. Worth noting, $\alpha$ is a penalty constant which was empirically set as 1.

As for the training process of the proposed network, it was very similar to that of conventional GAN. Specifically, the technique of back-propagation was utilized in the training process to minimize $L_d$ and $L_g$ alternately. When the network converged, it came to the detection phase. In this phase, the generator was removed and the discriminator was utilized as a detector to deal with the real spectra. Since we could utilize the discriminator at different times and with different weights, several initial detection maps could be acquired. To further suppress the background and the tiny outliers, necessary spatial processing techniques were adopted.

### 2.3.3. Bank of Guided Filters

Furthermore, since spatial correlation was useful in detecting certain objects in HSIs, a bank of guided filters could be utilized to smoothen the initial detection results. The framework of spatial filtering with guided filters is illustrated in Figure 2.

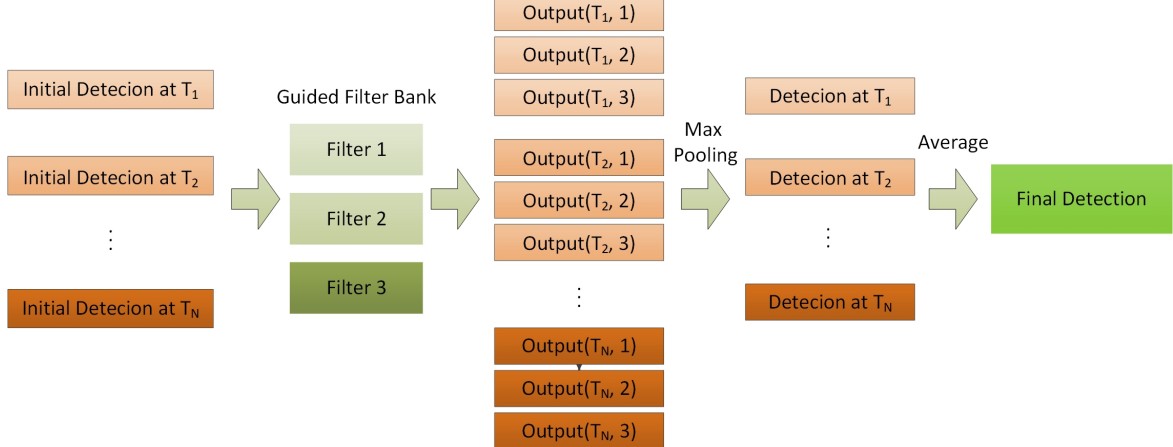

**Figure 2.** Spatial filtering with guided filters.

As shown in Figure 2, the initial detection maps obtained at different times were firstly filtered by the bank of filters individually. In this case, all the filters would adopt the initial input image as the guided image and they differed from each other only in radius *r* and the regularization coefficient $\epsilon$. Then, different degrees and scales of smoothening procedures were conducted on the initial detection maps. After that, max pooling was performed on the filtered images to gather the most valuable information. At last, an overall average value regarding the images acquired at different times was calculated so as to obtain a relatively stable detection result.

## 3. Results

To evaluate the robustness and effectiveness of the proposed approach, experiments were conducted for the target detection of HSIs. Firstly, the assessment criteria of target detection were provided along with the hyperspectral datasets that were utilized in this paper. Then, the preferences of the proposed network were analyzed through the experimental results. Finally, comparisons were performed between our approach and some other detection approaches such as ACE, CEM, STD, etc. The results showed the proposed approach could yield superior detection performance in terms of both efficiency and accuracy.

### 3.1. Assessment Criteria

In order to assess the detection performance reliably, a 2D receiver operating characteristic (ROC) curve was adopted together with the intuitional detection maps. As an effective metric for the ROC curve, the area under the curve (AUC) of the false positive rate ($P_f$) versus the true positive rate ($P_t$) was calculated. $P_f, P_t$ could be defined as Equation (9):

$$P_f = \frac{\text{FP}}{\text{FP} + \text{TN}}, P_t = \frac{\text{TP}}{\text{TP} + \text{FN}} \tag{9}$$

where FP and TP denote the false positive samples (backgrounds which were detected as targets) and the true positive samples (targets which were detected as targets), respectively, and FN and TN denote the false negative samples (targets which were detected as backgrounds) and the true negative samples (backgrounds which were detected as backgrounds), respectively. Therefore, $P_f$ was also known as the false alarm rate and $P_t$ the recall rate in the field of TD.

### 3.2. Datasets

Three HSI datasets were utilized in our experiments which were described as follows.

### 3.2.1. Airport–Beach–Urban Dataset

This dataset is available online and consists of scenes from airports, beaches and urban areas. Three images from each scene were selected for the following experiments, and were denoted as D1, D2 and D3, respectively. All of them were acquired by the airborne visible/infrared imaging spectrometer (AVIRIS) sensor and contained $100 \times 100$ pixels in the spatial domain. After removing the water-absorption and noisy bands, each group of HSIs included 205, 193, 207 bands, respectively. Their color composites and the corresponding reference maps are shown in Figure 3, and five target signatures were utilized as the training samples for each image.

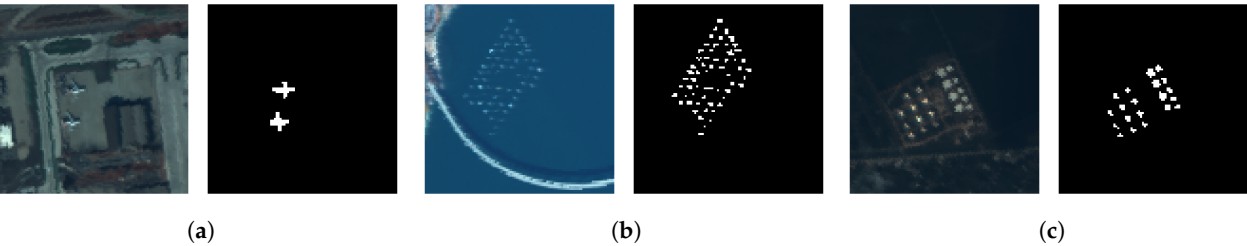

(a) (b) (c)

**Figure 3.** Color composites and reference maps: (**a**) D1, (**b**) D2 and (**c**) D3.

### 3.2.2. San Diego Airport Dataset

The second dataset was acquired by the AVIRIS sensor over the San Diego airport in CA, USA. The original HSIs contained $400 \times 400$ pixels in the spatial domain and 189 bands after removing the water-absorption and noisy bands. Two small parts were utilized for the experiments. They all contained $100 \times 100$ pixels and were denoted as D4 and D5, respectively. The color composites and the corresponding reference maps are illustrated in Figure 4a,b. Each of them contained three planes which were considered as the targets to be detected and five target signatures were selected as the training samples for each image.

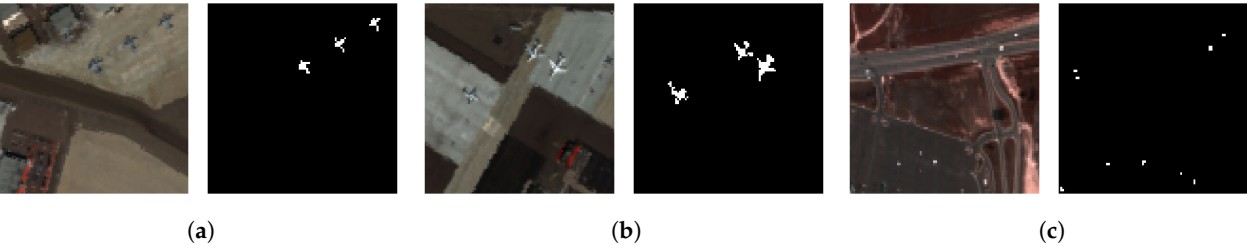

(a) (b) (c)

**Figure 4.** Color composites and reference maps: (**a**) D4, (**b**) D5 and (**c**) D6.

### 3.2.3. Urban Dataset

The third dataset was collected by the hyperspectral digital imagery collection experiment (HYDICE) sensor and its spatial and spectral resolutions were 2 m and 10 nm, respectively. A small part denoted as D6 with $80 \times 100$ pixels and 162 bands was utilized in the experiments. The color composites and the corresponding reference maps are shown in Figure 4c. There were 21 target pixels to be detected, where 3 target signatures were utilized as the training samples.

### 3.3. Preferences

Before further discussion, explanations were first determined here to parameter preferences, including the ratio between the target signature and the unlabeled spectra in one batch during training, the learning rate along with the number of classifiers and the guided filters during detection.

Earlier in this paper, we mentioned that the training set was composed of the target spectra and the unlabeled spectra. Since the former contained far fewer samples than the latter, only a small fraction of unlabeled spectra was selected as the backgrounds in each batch for training in the experiments. In this way, the imbalance between the two kinds of samples was greatly alleviated. In addition, the amount of target signature was doubled by adding Gaussian noises. Furthermore, the detection performance evaluated by the AUC $(P_f, P_t)$ under different ratios between the target spectra (including noisy samples) and the unlabeled spectra in one batch is shown in Figure 5a.

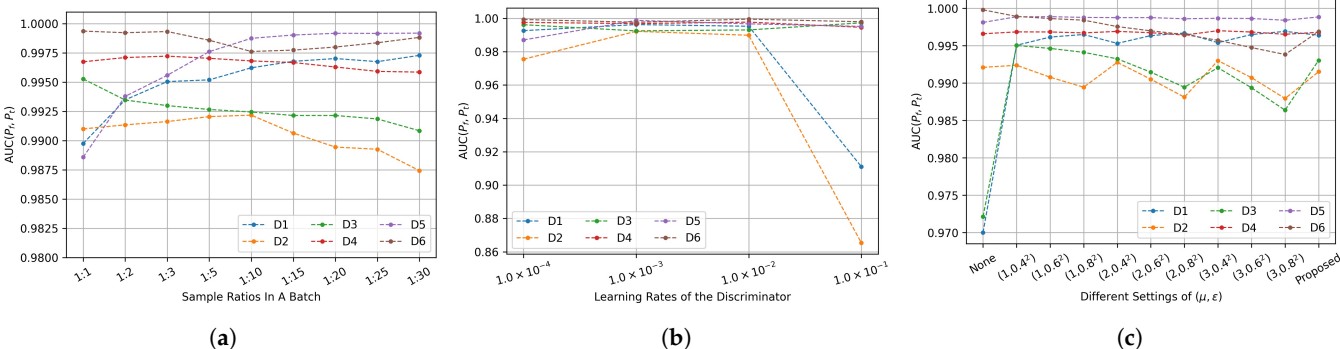

**Figure 5.** AUC $(P_f, P_t)$ under different (**a**) ratios between samples in one batch, (**b**) learning rates of the discriminator, (**c**) settings of guided filters.

From the results, we could see the ratio between the target spectra and the unlabeled spectra had an obvious influence on the detection performance, and the optimum ratio varied from different datasets. Specifically, for D1 and D5, as the proportion of unlabeled spectra in one batch increased, the AUC increased until it converged. For D3, D4 and D6, when the proportion of unlabeled spectra increased, the AUC decreased slightly. Furthermore, for D2, with the increasing of the proportion of unlabeled spectra, the AUC first increased slightly and then decreased dramatically. Judging from the different situations, we selected 1:10 as a proper ratio between the target spectra and the unlabeled spectra for the following experiments.

With respect to the learning rate, we adopted an imbalanced learning rate as in [28] with $l_d = [10^{-4}, 10^{-3}, 10^{-2}, 10^{-1}]$, $l_g = l_d/2$ for the discriminator and the generator, respectively, and employed RMSprop as the optimizer. Since we used spectral normalization and the smooth penalty to stabilize the training process, the network could converge in less than two thousand iterations. The detection performance evaluated by the AUC $(P_f, P_t)$ under different learning rates is shown in Figure 5b.

According to the results of Figure 5b, the learning rate had a significant influence on the detection performance, especially for D1 and D2. With a low degree of network convergence, a small learning rate such as $10^{-4}$ led to a decrease in the performance while a larger learning rate such as $10^{-1}$ could lead to a divergent network, resulting in a dramatic performance collapse. Meanwhile, the AUC changed slightly and exhibited a relatively robust performance when the learning rate fell into a range from $10^{-3}$ to $10^{-2}$. Given this fact, the learning rate of $10^{-3}$ was adopted in the following experiments.

After the network converged, it came to the detection phase. Discriminators at 15 different moments were utilized as the detectors, and 9 different guided filter preferences were selected to smoothen the initial detection maps. The pairs between the radius $r$ and regularization coefficient $\epsilon$ of guided filters were set to be $r = [1, 2, 3]$ and $\epsilon = [0.4^2, 0.6^2, 0.8^2]$. The detection performance before and after filtering with different filters is listed in Figure 5c. The proposed approach adopted three different settings for filters, viz. $(1, 0.8^2)$, $(2, 0.6^2)$ and $(3, 0.4^2)$, so as to form a filter bank as mentioned in the previous section.

As indicated in Figure 5c, the AUC ($P_f$, $P_t$) varied as the guided filter setting changed. The use of guided filtering improved the detection performance in almost every dataset, and the case was especially true for D1 and D3. However, the influence of filter settings on different datasets was quite different. For D1, (3, $0.8^2$) was the best setting for ($\mu$, $\epsilon$) of guided filters, while for D3, (1, $0.4^2$) was the optimal setting. Meanwhile, the AUC was almost the same for both D4 and D5. As the proposed approach adopted a bank of guided filters, it could always perform well in various datasets.

As neural networks on which the proposed approach was based on are often time-consuming with high computational complexity, the relevant processing complexity was explicitly addressed here to further demonstrate the effectiveness of the proposed approach. Specifically, the processing cost was evaluated in terms of the processing time (in seconds) and the amounts of learnable parameters, with results illustrated in Table 2. Meanwhile, the programs in the proposed approach were ran through Tensorflow in Python on a computer with a Quad-Core core processor of 3.40 GHz (i5-7500), GTX 1070 and 16G memory. As indicated in Table 2, a range of 0.15 to 0.17 million parameters (varying with the number of bands) was employed in the proposed network, the number of which was relatively small and was able to yield efficient detection results. Meanwhile, as the proposed approach was pixel-based, the processing time would change as the size of datasets changed. For this reason, D6 (with a smaller size of 80 × 100) showed a better detection efficiency than the other datasets.

**Table 2.** The processing cost of the proposed approach in various datasets.

| Datasets | | D1 | D2 | D3 | D4 | D5 | D6 |
|---|---|---|---|---|---|---|---|
| Time (sec.) | Train | 11.83 | 12.25 | 12.06 | 12.03 | 11.89 | 9.57 |
| | Test | 3.05 | 3.12 | 3.06 | 2.99 | 3.08 | 2.90 |
| Learnable Parameters | | $1.5 \times 10^5 \sim 1.7 \times 10^5$ | | | | | |

### 3.4. Comparisons with Other Approaches

In this paper, we compared the proposed approach with two traditional approaches (i.e., ACE [1] and CEM [6]), two sparse-based approaches (i.e., sparse representation for target detection (STD) [7], and combined sparse and collaborative representation for target detection (CSCR) [29]) and a neural network-based approach (i.e., semi-supervised classification based on GAN (SSC) [17]). A global background estimation strategy was utilized in ACE, while an adaptive local dual window approach was adopted for the background dictionary estimation as in [29]. Furthermore, in consideration of the varied sizes of different targets, dual windows with various sizes were adopted for different datasets. For D3, the dual window size was [3, 17], while for D6, the dual window size was set to be [5, 9]. Moreover, the dual window sizes for the remaining datasets were set as [5, 13]. The detection performance is shown in Table 3, Figures 6 and 7.

Table 3 shows the detection performance evaluated by AUC ($P_f$, $P_t$). From this table, we could see the proposed approach performed well and was able to achieve the highest AUC ($P_f$, $P_t$) in almost every dataset except for D3 (the result of which was only 0.0017 lower than the maximum value). The other approaches performed well in certain datasets, but +are likely to fall behind the proposed approach when dealing with other datasets. Furthermore, it is noteworthy that our approach was able to outperform the approach of SSC (which bears strong resemblance to our approach) in almost every dataset, especially in D2, which further validated the effectiveness of the unified training process with a composite loss function. Based on this fact as well as the other facts that were elaborated earlier, we could draw the conclusion that the proposed approach was more robust and could deal with various sorts of HSIs without resetting the hyperparameters.

Figure 6 shows the comparison results between different algorithms in a more visual way. The performance of a specific approach was displayed in one column. From this figure, we could observe that the detection results were quite in accord with what was shown in Table 3, which also supported the conclusion that the proposed approach was more robust when dealing with various HSIs. Furthermore, with the use of guided filters, the contour edges were preserved to a considerable extent.

**Table 3.** Detection results evaluated by AUC $(P_f, P_t)$ of different approaches.

| Dataset | AUC | ACE | CEM | STD | CSCR | SSC | Proposed |
|---------|-----|-----|-----|-----|------|-----|----------|
| D1 | $(P_f, P_t)$ | 0.9155 | 0.9199 | 0.8854 | 0.9163 | 0.9891 | **0.9962** |
| D2 | $(P_f, P_t)$ | 0.8786 | 0.8840 | 0.9130 | 0.9581 | 0.9420 | **0.9922** |
| D3 | $(P_f, P_t)$ | 0.8530 | 0.8160 | 0.9421 | 0.9717 | **0.9941** | 0.9924 |
| D4 | $(P_f, P_t)$ | 0.9248 | 0.9163 | 0.8883 | 0.9538 | 0.9884 | **0.9968** |
| D5 | $(P_f, P_t)$ | 0.9953 | 0.9965 | 0.9966 | 0.9979 | 0.9929 | **0.9987** |
| D6 | $(P_f, P_t)$ | 0.9789 | 0.9895 | 0.9962 | 0.9968 | 0.9974 | **0.9976** |

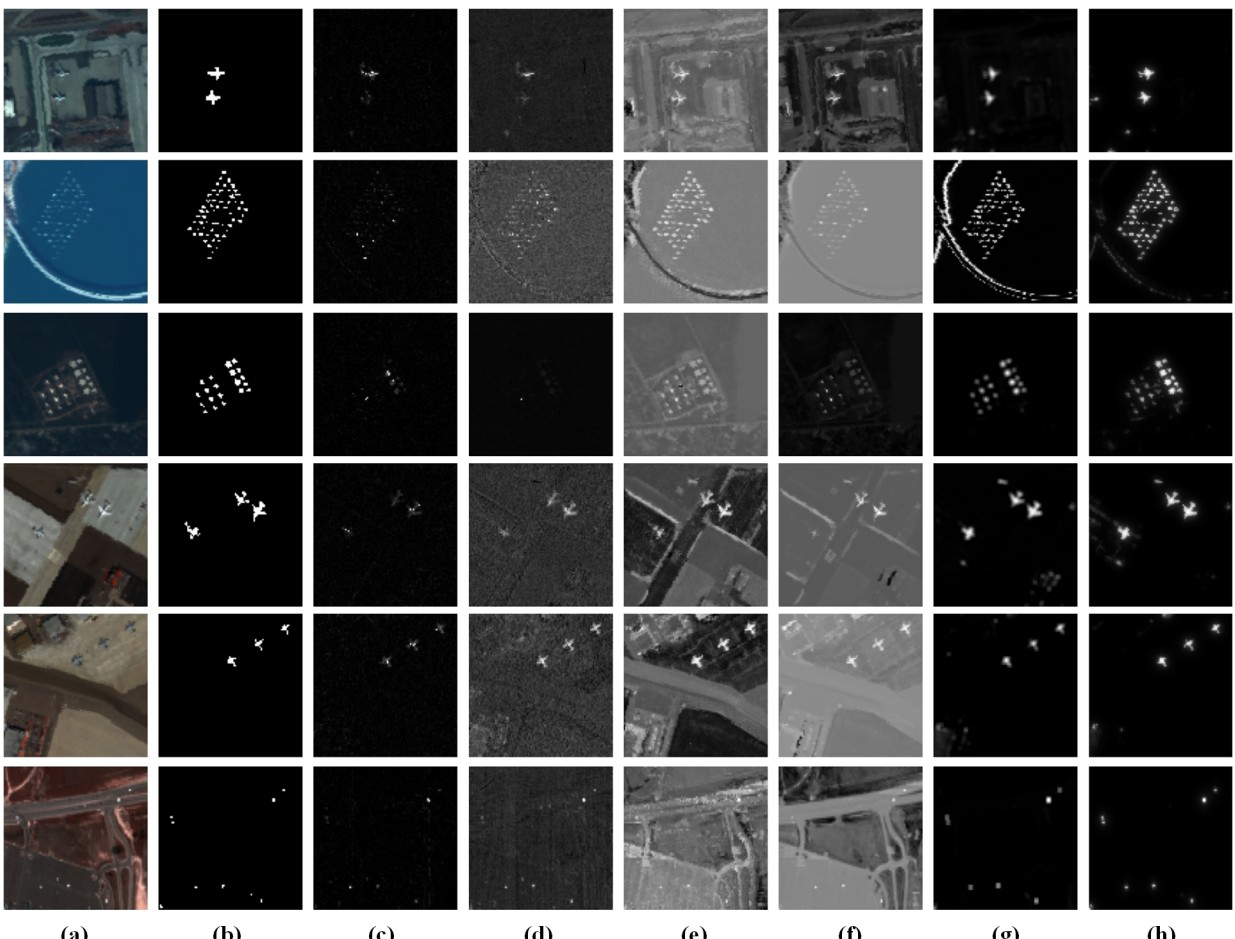

**Figure 6.** Detection results of different approaches: (**a**) color composites, (**b**) reference maps, (**c**) ACE, (**d**) CEM, (**e**) STD, (**f**) CSCR, (**g**) SSC and (**h**) the proposed.

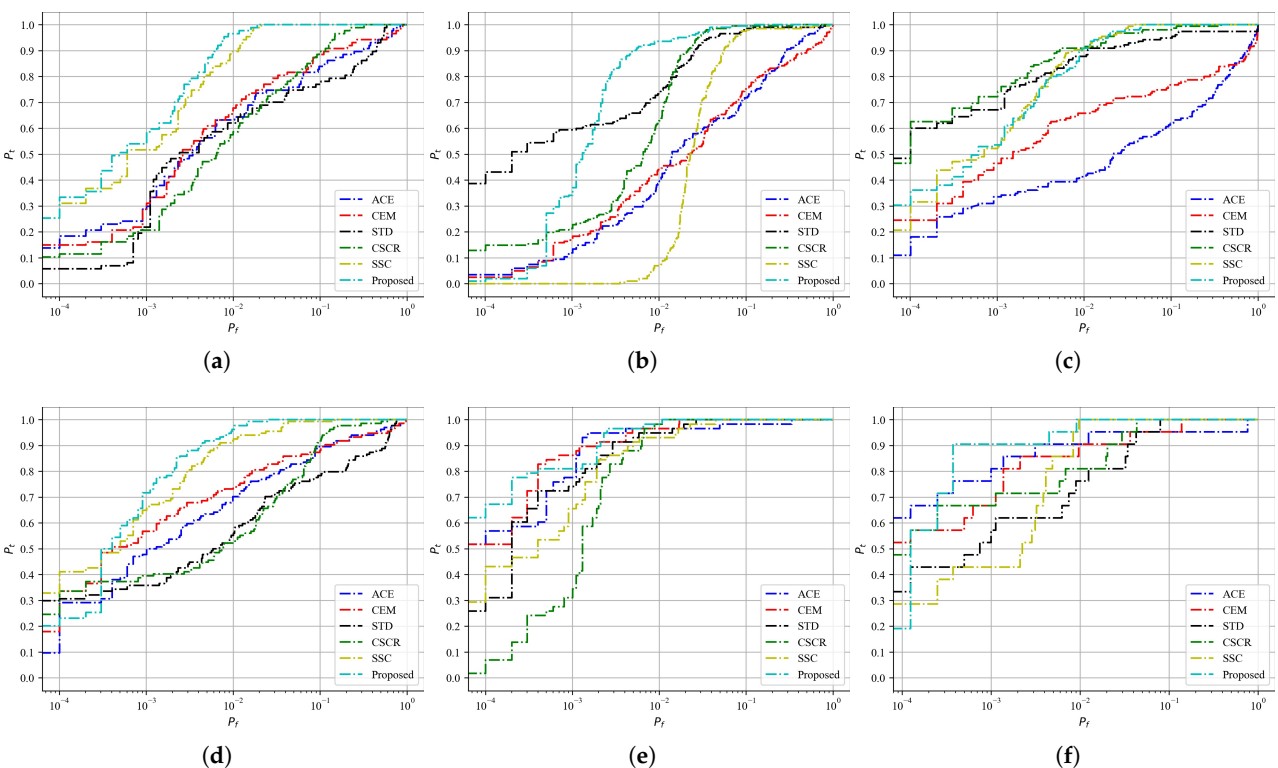

**Figure 7.** ROC curve ($P_f$, $P_t$) of different datasets: (**a**) D1, (**b**) D2, (**c**) D3, (**d**) D4, (**e**) D5 and (**f**) D6.

Besides what was mentioned above, additional points that could be determined from the ROC curve of Figure 7 were: for D1 and D4, the proposed approach outperformed the other approaches under various false positive rates (also known as false alarm rates) and was able to allow $P_t$ to reach its peak value at a very small false alarm rate. Meanwhile, for D5 and D6, the gap between the proposed approach and the other approaches was very small when the false alarm rate was bigger than $10^{-2}$. Furthermore, the former displayed a more competitive performance when the false alarm rate dropped to a value below $10^{-3}$. With regard to D2 and D3, the situation became more complex. When the false alarm rate dropped to a low value, $P_t$ in certain approaches was higher. However, when the false alarm rate was higher than a certain value (about $2 \times 10^{-3}$ for D2, $10^{-2}$ for D3), $P_t$ of the proposed approach outperformed the other approaches and quickly reached its maximum point. Such property of the proposed approach would make a much higher AUC as shown in Table 3.

## 4. Discussion

It was found that similarities exist between and among approaches of [17,18] and our approach as in these approaches, the technique of adversarial training process was exploited to fix the problem of limited labeled samples. However, it should be noted that in [17,18], the training process was separated into different phases, which was likely to result in a weakened correlation between networks. To avoid the gap between the different training phases, a neural network with a composite loss function was built in our approach, which was able to crack the problem to a comparatively satisfactory extent.

Another point that may need to be addressed more clearly here is the proposed approach utilized generated samples in the training process, the technique of which usually falls under data augmentation. However, visible differences were present between the proposed approach and other augmentation approaches such as [30,31]. Specifically, the two processes, i.e., data augmentation and the training of the target detector, were realized simultaneously as a whole in the proposed approach, but were realized sequentially in the others. Furthermore, more importantly, as a data-driven approach, no prior assumptions

were needed for GAN in the proposed approach, whereas for other approaches, hypotheses (e.g., linear mixing model or subspace) are often needed for data augmentation.

Alongside the efforts performed in the proposed approach, further research can be conducted on how to develop an approach which is entirely based on neural networks for target detection using a single-target signature.

## 5. Conclusions

In sum, we proposed a novel hyperspectral target detection framework with an auxiliary generative adversarial network in this paper. Simulated samples of target and background spectra were generated by the generative neural network to facilitate the target detection process. Extensive experiments conducted on the HSI datasets showed the proposed target detection approach equipped with the post-processing technique of guided filtering was more competitive when compared to other approaches.

**Author Contributions:** Y.G. and Y.F. conceived and designed the study. Y.G. constructed the neural network, implemented the experiments and drafted the manuscript. X.Y. contributed to the improvement of the model and the manuscript. Y.F. provided the overall guidance to the project, reviewed and edited the manuscript. All authors have read and agreed to the published version of the manuscript.

**Funding:** This research is funded by Shaanxi Key Industry Innovation Chain Project (Grant Number: 2019ZDLGY14-02-02).

**Data Availability Statement:** Publicly available datasets were analyzed in this study. The datasets can be found here: http://levir.buaa.edu.cn/code_cn.htm, https://www.erdc.usace.army.mil/Media/Fact-Sheets/Fact-Sheet-Article-View/Article/610433/hypercube/ and http://xudongkang.weebly.com/data-sets.html, accessed on 3 November 2021.

**Conflicts of Interest:** The authors declare no conflict of interest.

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
