# Peer review of "Hyperspectral Target Detection with an Auxiliary Generative Adversarial Network"

_remotesensing, doi:10.3390/rs13214454_

Round 1
Reviewer 1 Report
- The Authors in the article presented a novel hyperspectral target detection approach with an auxiliary generative adversarial network. The proposed network consists of two parts: a generator and a discriminator. The generator generates the simulated target spectra and background spectra. This simulated data is then acquired by a classifier which is highly correlated with the discriminator. The obtained numerical results showed the advantages of proposed network model: the smaller processing cost and higher true detection results comparing with five different approaches.
- In Introduction the Authors make revision of world bibliography concerning different methods of hyperspectral images detection, applications of deep neural networks, feature extraction and ways of generation target samples needed for training neural network.
- In my opinion, the problem of insufficient training samples also occurs in detection and recognition another kinds of targets and objects, therefore the following articles concerning images recognition based on the measured data of their parameters with using neural networks are also supposed to be listed in the References:
- Chen, Hongyi; Zhang, Fan; Tang, Bo; et al., Slim and Efficient Neural Network Design for Recourse-Constrained SAR Target Recognition, REMOTE SENSING, 10, Issue: 10, Article Number: 1618, Published: OCT 2018.
- Matuszewski, J.; Pietrow, D. Recognition of electromagnetic sources with the use of deep neural networks. XII Conference on Reconnaissance and Electronic Warfare Systems, Ołtarzew, Poland, 19-21.11.2018, SPIE 11055, 110550D. 2019, DOI: 10.1117/12.2524536.
- Xu, Wang; Chen, Renwen; Huang, Bin; et al., Single Image Super-Resolution Based on Global Dense Feature Fusion Convolutional Network, SENSORS, : 19, Issue: 2, Article Number: 316, Published: JAN 2 2019.
- The architecture of proposed model of deep neural network for target detection is correctly depicted in the Figure 1 and in Table 1.
- The successive steps of data generation, testing and validation of proposed target detection method with appropriate mathematical formulas and computing examples are correctly described in Section 2. All presented in this article mathematical formulas and used symbols are well defined and edited.
- The calculation results are shown in the Tables 2-3 and depicted in Figures 5-7 with appropriate comments.
- Remarks:
- In which units the cost of calculations is given in Table 2 (seconds, minutes ?)
- Does the false and true positive ratio in formula (9) mean the same as the probability of correct and incorrect target detection.
- The abbreviations InfoGAN and ACGAN in line 176) are not deciphered.
- The simulation results confirmed the usefulness of proposed method by the Authors for dataset taken to calculations. The proposed by the Authors model of deep neural network improve the operation speed and effectiveness of images detection.

Author Response
The authors would like to thank the reviewer for his/her constructive comments and suggestions which would help us to improve the quality of the paper.
Please see the attachment for an item-by-item response.

Reviewer 2 Report
The authors propose to minimize the class imbalance of target materials with GAN. Such implementation falls under data augmentation. GAN is a common technique in augmentation, among others. As such, the authors must differentiate their work from papers like https://doi.org/10.1109/ICIST.2019.8836913 and https://doi.org/10.1016/j.patcog.2020.107464, which is the primary correction to address. Aside from that, the paper has minor issues like moving the ROC curve figure (figure 7) from the reference section.
Author Response
Response to Review 2 Comments
The authors would like to thank the reviewer for his/her constructive comments and suggestions which would help us to improve the quality of the paper. We have revised the paper according to the reviewer’s comments, fixed some small spelling or written errors and modified several sentences for clear explanation.
An item-by-item response to the following remarks are as follows.
Point 1: The authors propose to minimize the class imbalance of target materials with GAN. Such implementation falls under data augmentation. GAN is a common technique in augmentation, among others. As such, the authors must differentiate their work from papers like https://doi.org/10.1109/ICIST.2019.8836913 and https://doi.org/10.1016/j.patcog.2020.107464, which is the primary correction to address.
Response 1: Thank the reviewer for the comments and providing the references for comparison. We have added a short paragraph from line 449 to line 457 in Section 4 to discuss the differences between the proposed approach and other techniques in data augmentation. And the relevant modifications are also included here for your convenient reference:
“Another point that may need to be addressed more clearly here is the proposed approach utilizes generated samples in the training process, the technique of which usually falls under data augmentation. But visible differences are present between the proposed approach and other augmentation approaches such as [30] and [31]. Specifically, the two processes, i.e. data augmentation and training of the target detector, are realized simultaneously as a whole in the proposed approach, but are realized sequentially in the others. And more importantly, as a data-driven approach, no prior assumptions are needed for GAN in the proposed approach, whereas for other approaches, hypotheses (e.g. linear mixing model or subspace) are often needed for data augmentation.”
Point 2: Aside from that, the paper has minor issues like moving the ROC curve figure (figure 7) from the reference section.
Response 2: We have checked the whole paper and modified the improper position of Figure 7. Thank the reviewer for the comments which enable us to finely tune the paper to a more satisfactory degree.

Round 2
Reviewer 2 Report
The authors have performed the necessary changes requested. I recommend acceptance of this manuscript, pending any English and editorial corrections. Good luck!